# Diagnostic Accuracy of Liquid Biomarkers for the Non-Invasive Diagnosis of Endometrial Cancer: A Systematic Review and Meta-Analysis

**DOI:** 10.3390/cancers14194666

**Published:** 2022-09-25

**Authors:** Rebecca Karkia, Sarah Wali, Annette Payne, Emmanouil Karteris, Jayanta Chatterjee

**Affiliations:** 1Academic Department of Gynaecological Oncology, Royal Surrey NHS Foundation Trust, Surrey, Guildford GU2 7XX, UK; 2Brunel Department of Life Sciences, Brunel University London, Kingston Lane Uxbridge, Middlesex, Uxbridge UB8 3PH, UK; 3Department of Obstetrics and Gynaecology, Chelsea & Westminster NHS Foundation Trust, 369 Fulham Road, London SW10 9NH, UK; 4Brunel Department of Computational Science, Brunel University London, Kingston Lane Uxbridge, Middlesex, Uxbridge UB8 3PH, UK

**Keywords:** endometrial cancer, liquid biopsy, non-invasive biopsy, liquid biomarkers

## Abstract

**Simple Summary:**

Endometrial cancer is common amongst women and rates are increasing annually. The diagnosis of this condition for women with bleeding after the menopause is invasive and often painful with many more women undergoing investigation than needed. A simple, non-invasive blood or urine test for the diagnosis of endometrial cancer is being sought. This review summarizes the current research on blood and urine tests and their diagnostic accuracy for detecting endometrial cancer. Whilst many blood and urine tests have been assessed there is currently no test that has a similar accuracy to biopsy of the uterine lining. However, this review demonstrates that there are some potential candidates which need to be explored by larger studies and on bigger groups of women.

**Abstract:**

Endometrial cancer rates are increasing annually due to an aging population and rising rates of obesity. Currently there is no widely available, accurate, non-invasive test that can be used to triage women for diagnostic biopsy whilst safely reassuring healthy women without the need for invasive assessment. The aim of this systematic review and meta-analysis is to evaluate studies assessing blood and urine-based biomarkers as a replacement test for endometrial biopsy or as a triage test in symptomatic women. For each primary study, the diagnostic accuracy of different biomarkers was assessed by sensitivity, specificity, likelihood ratio and area under ROC curve. Forest plots of summary statistics were constructed for biomarkers which were assessed by multiple studies using data from a random-effect models. All but one study was of blood-based biomarkers. In total, 15 studies reported 29 different exosomal biomarkers; 34 studies reported 47 different proteomic biomarkers. Summary statistic meta-analysis was reported for micro-RNAs, cancer antigens, hormones, and other proteomic markers. Metabolites and circulating tumor materials were also summarized. For the majority of biomarkers, no meta-analysis was possible. There was a low number of small, heterogeneous studies for the majority of evaluated index tests. This may undermine the reliability of summary estimates from the meta-analyses. At present there is no liquid biopsy that is ready to be used as a replacement test for endometrial biopsy. However, to the best of our knowledge this is the first study to report and meta-analyze the diagnostic accuracy of different classes of blood and urine biomarkers for detection of endometrial cancer. This review may thus provide a reference guide for those wishing to explore candidate biomarkers for further research.

## 1. Introduction

Endometrial cancer (EC) is the most common form of uterine cancer and arises from the lining of the uterus, known as the endometrium. In 2020, GLOBOCAN the World Health Organizations’ International Agency for Research on Cancer recorded 417,367 new cases of uterine cancer and 97,370 deaths making this the fourth most common cancer of women and the third leading cause of cancer mortality in females after breast and lung cancer [1]. Incidence in the UK has increased by around 55% since the 1990s with 9700 women diagnosed per year in 2016–2018 [2]. The rise in cases of EC diagnosed annually is set to increase globally by 30% by the year 2040 [3]. This is likely due to the advancing age of the population and rising rates of obesity. Currently if diagnosed and treated at International Federation of Gynecology and Obstetrics [FIGO] Stage I or II, EC 5-year survival rates are around ~92% and 75%, respectively, whereas advanced stage III and IV ECs have 5-year survival rates of 48% and 15%, respectively [4,5,6]. Strategies for early diagnosis are therefore critical. 

Symptoms suspicious for EC are post-menopausal bleeding, unscheduled bleeding on hormone replacement therapy (HRT), persistent intermenstrual or irregular bleeding, hematuria or abnormal vaginal discharge. Although EC is less commonly diagnosed in pre-menopausal women, in the UK, 6.5% of women diagnosed with EC between 2015–2017 were less than 50 years old [7]. In post-menopausal women with bleeding, transvaginal ultrasound scan (TVS) is performed as a triaging test to assess endometrial thickness (ET). An ET < 4 mm is associated with an <1% risk of EC [8,9]. For those with an ET > 4 mm, an endometrial biopsy is recommended as the gold standard diagnosis of EC [8]. However, in symptomatic perimenopausal or pre-menopausal women with risk factors for EC, the National Institute of Clinical Excellence recommend hysteroscopy and targeted biopsy in the first instance as TVS has limited value in women who are still menstruating [10].

Whilst the gold standard of EC diagnosis, endometrial biopsy has an almost 99% accuracy, the current technique for triaging with ultrasound lacks specificity resulting in more than 50% of patients needing invasive biopsy [11]. Furthermore, failure to obtain a biopsy in the outpatient setting is common and occurs in around one third of women, often due to sampling failure or pain during the investigation [12]. In these patients, repeat investigations are needed, often under general anesthetic which is not without associated risks and costs [13]. A simple, easy to administer, non-invasive test that could triage women with EC for diagnostic biopsy whilst safely reassuring healthy women, would vastly improve patient care over the current model. The ideal detection tool would be simple to perform, non-invasive to obtain, accurate in reassuring women without disease and cost effective to allow implementation as a screening program within the primary care setting. 

‘Liquid biopsy’ has the potential to offer this solution and refers to the sampling and analysis of non-solid biological tissue for its tumoral elements [14]. These elements might be circulating tumor cells (CTCs), cell-free tumor DNA (ctDNA), as well as extracellular vesicles (EVs), microRNAs (miRNAs), mRNA, long noncoding RNAs (lncRNAs), small RNA, circulating cell-free proteins, and tumor-educated platelets (TEPs) [14]. Many body fluids can be used for liquid biopsy, however the most non-invasive and widely investigated fluids are blood and urine. Multiple approaches have been employed in the search for diagnostic cancer biomarkers. Advances in areas such as proteomics, metabolomics and genomic sequencing have increased the scope for individual or panels of biomarkers to be discovered. However, biomarkers must overcome several hurdles before they are implemented into clinical practice; discovery, validation, and verification [14]. An ideal biomarker should be accurate and reproducible between laboratories. Its clinical validity should be reported in terms of sensitivity, specificity, positive predictive value (PPV) and negative predictive value (NPV). However, between given studies, it is difficult to compare the performance of one or multiple biomarkers unless they share a fixed false positive rate. For this reason, the area under the curve (AUC) value of a receiver operator characteristic (ROC) curve plotting Sensitivity over the False Positive Rate (1- Specificity) gives an objective comparison between test performance with one value and is therefore the best tool of comparison between studies analyzing the same marker [15,16].

The aim of this systematic review is to evaluate studies assessing blood and urine-based biomarkers as a replacement test for endometrial biopsy or as a triage test to inform the decision to perform endometrial biopsy. Specific objectives are to provide summary estimates of the diagnostic accuracy of blood or urinary biomarkers for the diagnosis of EC compared to endometrial biopsy and to assess the diagnostic utility of biomarkers that could differentiate between benign and malignant endometrium. 

## 2. Materials and Methods

The study was carried out according to the Cochrane Collaboration recommendations as well as the Preferred Reporting Items for Systematic Review and Meta-analysis of Diagnostic Accuracy Studies (PRISMA-DTA) [17]. This study was registered with PROSPERO, registration number: CRD42020202191 [18]. 

### 2.1. Literature Search

Systematic literature searches were carried out in the NICE Healthcare Databases Search tool using CINAHL, EMBASE, Medline and Pubmed. The database was searched from 2000 to January 2022. The search was restricted to English language papers, full text articles and non- review articles. The search strategy included the following key words: (endometrial cancer [Title/Abstract]) AND (biomarker [Title/Abstract]. Cross-referencing of key texts and grey literature (Google Scholar) searches were also carried out.

Titles and abstracts were screened for the eligibility of the study and evaluated by two different operators (RK, SW). Relevant articles were retrieved in full-text and assessed against the inclusion criteria. Inclusion criteria were as follows; firstly, studies evaluating the diagnostic accuracy of either blood or urine biomarkers for EC detection; second, those studies diagnosing EC using endometrial histopathological assessment as the reference test; third, those studies reporting AUC as well as sensitivity and specificity and fourth those studies comparing a control group cohort without EC. Reviews, letters, conference reports, and duplicated publications were excluded to make sure only primary publications of original studies were included. Studies not reporting diagnostic accuracy with AUC and studies where biomarker performance was reported with only grouped markers were excluded from further analysis. Studies solely reporting HE4 or CA125 where multiple systematic reviews and meta-analyses have already been published were excluded from further review [19,20,21,22]. 

### 2.2. Data Extraction and Quality Assessment

Data was extracted using a standard form that included methods, basic study population characteristics, inclusion and exclusion criteria and accuracy of diagnostic tests used.

The risk of bias was evaluated by three independent authors (RK, SW, JC) in each study using the Diagnostic Precision Study Quality Assessment Tool (QUADAS-2) as recommended by the Cochrane Collaboration [23]. Any discrepancies between authors were discussed and the opinion of a third reviewer was sought. The assessment was carried out with use of the Review Manager Software (Version 5.4, The Cochrane Collaboration, London, UK, 2020).

Each study was deemed to be at ‘low’, ‘high’ or ‘unclear’ risk for each of four domains. Studies were deemed as having low methodological quality when they were at high or unclear risk of bias or when there was high concern regarding applicability in at least in one domain. The original signaling question, ‘Was a case-control design avoided?’ was amended to ‘Was a two-gate design avoided?’ in agreement with the Cochrane study of Nissenblat et al., 2016 [24]. Diagnostic accuracy studies are cross-sectional in nature, comparing an index test with the reference standard in the same group of participants. Study investigators measure the parameters at a single point in time and classify the groups by the outcome of the reference standard test. Therefore, unlike in epidemiological studies, the terminology ‘cohort’ and ‘case- control’ is less informative so ‘single-gate’ and ‘two-gate’ design was subtitled. This question was included because a two-gate design has more potential to introduce selection bias and overestimate accuracy of a given biomarker [25]. 

### 2.3. Statistical Analysis

For each primary study, a diagnostic accuracy of different biomarkers was assessed by sensitivity, specificity, likelihood ratios and area under ROC curve with their 95% confidence intervals. It is common that primary studies report sensitivity and specificity for a study-specific cut off. AUC cannot be generated where source data is not available and thus it was necessary for inclusion criteria for this to be reported. Where the AUC 95% confidence interval (CI) was not reported in the text, study authors were contacted individually for missing data. Where no response was received the decision to calculate the standard error of the mean using non-parametric Wilcoxon’s statistics was assessed individually [15,26]. If study numbers of cases and control patients reached less than 60 and no 95% CI for AUC was reported, the reliability of this method was deemed suboptimal and hence the paper was excluded from meta-analysis. A weighted summary AUC was calculated with the assumption of non-homogeneity and non-normality of empirical sensitivity and specificity as reported by Zhou et al. [15]. Forest plots of summary statistics were constructed using the data from the random-effect models. The heterogeneity of the studies was established by using Cochran’s heterogeneity statistics Q, which was calculated as the weighted sum of squared differences between the individual study effects and pooled effect across the studies [27]. I^2^, assesses the percentage of variation across individual studies that is due to heterogeneity rather than chance and does not depend on the number of studies included in the meta-analysis [27]. Values (I^2^) of 0–40% might be insignificant, 30–60% considered as moderate, 50–90% substantial and 75–100% considerable. Publication bias was evaluated by the Egger regression and Beggs’ correlation tests with funnel plots [27]. Analyses were carried out with MedCalc software (MedCalc Software 20.112, Ostend, Belgium).

## 3. Results

The literature search of MEDLINE, EMBASE and PUBMED yielded 1951 citations. A total of 56 studies were included in the systematic review. Publications were then grouped by biomarker category into publications relating to exosomal biomarkers, proteomic biomarkers, metabolomic biomarkers and circulating tumor materials (CTMs). Proteomic biomarkers were further sub-categorized. Those biomarkers that were reported in studies on two or more occasions were included for meta-analysis. Those biomarkers reported by one study only were included in the systematic review and tabulated in order of test performance statistics. A flow diagram of the literature review is shown in Figure 1 in accordance with PRISMA guidance [28].

### 3.1. Assessment of Quality and Heterogeneity of Studies 

The methodological quality of studies was assessed using the QUADAS-2 for exosomal, proteomic, metabolomic and studies discussing circulating tumor material. Data was inputted and tabulated graphically as seen in Figure 2, Figure 3, Figure 4 and Figure 5. The key area of concern for multiple studies was the patient selection domain where a two-gate study design was chosen. 

### 3.2. Meta-Analysis of Exosomal Biomarkers

15 studies reported 29 different exosomal biomarkers and studied a total of 5527 patients of which 2530 had a diagnosis of EC and 2456 were non-cancerous control group patients. All biomarkers were obtained by blood sample. The control groups consisted of healthy women with normal endometrium as well as those with benign endometrial lesions such as polyps and fibroids. Only studies of micro-RNA 21, 27a, and 223 were suitable for meta-analysis after meeting the inclusion criteria [29,30,31,32,33,34,35]. None of the included studies reported positive predictive values (PPV) or negative predictive values (NPV) (Table 1). 

#### 3.2.1. Micro RNA-21 

Three studies examined the diagnostic accuracy of MiRNA 21 as a biomarker for the diagnosis of EC (Figure 5, Table 2). The study by Gao et al., 2016 examined test performance against two control cohorts, one with healthy endometrium and one cohort with benign endometrial changes [29]. Sensitivity and specificity varied between the studies (0.640–0.850 and 0.760–0.920, respectively). The summary weighted AUC was deemed excellent at 0.825 (95%CI 0.735–0.915, *p* < 0.001). There was considerable heterogeneity between studies as seen by a Q-test score of 13.4 and I^2^ of 77.6%.

#### 3.2.2. Micro-RNA 27a 

Mi-RNA 27a was assessed by two studies, Wang et al., 2014 and Ghazala et al., 2021 (Table 3) [32,33]. The study by Wang examined test performance against a control cohort with both normal and benign endometrial changes. The study by Ghazala included a control cohort with normal endometrium only. The summary weighted AUC was outstanding and calculated at 0.925 (95%CI 0.801–1.000, *p* < 0.001) (Figure 5). There was considerable heterogeneity between studies as seen by both the Q test and I^2^ test (10.3 and 90.3%, respectively).

#### 3.2.3. Micro-RNA 223 

Mi-RNA 223 was reported by three studies (Table 4) [32,34,35]. The summary weighted AUC for Mi-RNA 223 was excellent at 0.813 (95%CI 0.735 to 0.890, *p* < 0.001). Tests for heterogeneity showed moderate heterogeneity between studies (Q = 4.12, I^2^ = 51.5%). 

### 3.3. Protein Based Biomarkers 

There were 35 studies reporting 47 different proteomic biomarkers (after the exclusion of CA-125 and HE-4 as previously described) and a total of 3526 patients of which 1483 had a diagnosis of EC and 2043 were non-cancerous control group patients [44,45,46,47,48,49,50,51,52,53,54,55,56,57,58,59,60,61,62,63,64,65,66,67,68,69,70,71,72,73,74,75,76,77,78,79]. A total of 34 studies were assessing blood-based biomarkers and one study a urine derived biomarker [64]. The control groups comprised of healthy women with normal endometrium as well as those with benign endometrial lesions such as polyps and fibroids. The summary of the diagnostic accuracy and performance of those biomarkers is outlined in Table 5. For the purpose of meta-analysis, proteomic markers were classed as cancer antigens (CA-15.3, CA-19.9, CA-72.4, CEA), hormones (leptin, visfatin, prolactin) and other proteomic markers (G-CSF, YKL-40 and DJ-1).

#### 3.3.1. Cancer Antigens—CA 15-3

CA 15-3 was reported by three studies (Table 6) [48,55,70]. In the study of Unuvar et al., 2020, the diagnostic accuracy of CA 15-3 was assessed against healthy controls with normal endometrium as well as those with benign endometrial changes [48]. The summary weighted AUC showed good performance reporting 0.608 (95%CI 0.536–0.681 *p* < 0.001) (Figure 6). Tests for heterogeneity between studies showed insignificant heterogeneity (Q = 1.271, I^2^ = 0.00%; 95%CI: 0.00–69.53). 

#### 3.3.2. Cancer Antigen—CA 19-9

CA 19-9 was reported by five studies (Table 7) [48,54,55,67,71]. There was a lot of variation in reporting sensitivity and specificity amongst the studies (0.290 to 0.945 and 0.047 to 1.000, respectively) with Bian et al., 2017 not reporting sensitivity for any markers [71]. The summary weighted AUC for CA 19-9 was acceptable and calculated at 0.621 (95%CI 0.539 to 0.702, *p* < 0.001) (Figure 6). Cochrane Q and I2 tests showed considerable heterogeneity between studies (Q = 13.51, I^2^ = 85.19%; 95%CI: 56.38–94.97). 

#### 3.3.3. Cancer Antigen—CA 72-4

CA 72-4 was reported by three studies (Table 8) [45,71,72]. The summary weighted AUC for CA 72-4 was good at 0.666 (95%CI 0.488 to 0.845, *p* < 0.001) (Figure 6). There was significant heterogeneity between studies (Q = 53.99, I^2^ =96.30%; 95%CI: 92.19–98.24). 

#### 3.3.4. Cancer Embryonic Antigen (CEA) 

CEA was reported by five studies (Table 9) [48,55,67,70,73]. The summary weighted AUC for CEA was acceptable, at 0.607 (95%CI 0.542 to 0.671, *p* < 0.001) (Figure 6). There was moderate heterogeneity between studies (Q = 6.969, I^2^ = 42.60%; 95%CI: 0.00–78.90). There was marked inconsistency in reported sensitivity and specificity between the studies (0.342 to 0.882 and 0.427 to 0.950, respectively).

### 3.4. Homonal Biomarkers

#### 3.4.1. Leptin

Leptin was reported by three studies (Table 10) [50,51,53]. The summary weighted AUC for leptin was good at 0.757 (95%CI 0.882 to 0.531, *p* < 0.001) (Figure 7). There was moderate to substantial heterogeneity between studies as shown by Cochran Q (Q = 5.0851) and I^2^ statistics, I^2^ = 60.67% (95%CI: 0.00–88.79). 

#### 3.4.2. Prolactin 

Prolactin was reported by two studies (Table 11) [70,74]. The summary weighted AUC for Prolactin was excellent at 0.826, unfortunately however with wide confidence interval (95%CI 0.576 to 1.000, *p* < 0.001) as shown in Figure 7. There was considerable heterogeneity between studies (Q = 33.036, I^2^ = 96.98%; 95%CI: 92.09–98.84). There was also considerable disparity in reported sensitivity between the studies (0.386 and 0.983).

#### 3.4.3. Visfatin 

Visfatin was reported by two studies (Table 12) [61,76]. The summary weighted AUC for Visfatin was poor, 0.552 (95%CI 0.471 to 0.633, *p* < 0.001) (Figure 7). Additionally, there was a substantial heterogeneity between studies (Q = 3.683, I^2^ = 72.85%; 95%CI: 0.00–93.89). 

### 3.5. Other Proteomic Markers 

#### 3.5.1. YKL-40

YKL-40 was reported by six studies (Table 13) [48,72,76,77,78,79]. The summary weighted AUC for YKL-40 was good, calculated at 0.757 (95% CI 0.667–0.848, *p* < 0.001) (Figure 8) [48,70,74,75,76,77]. There was a considerable heterogeneity amongst the studies (Q = 23.8, I^2^ = 78.98%; 95%CI: 53.94–90.41). There was a significant variation in reported sensitivity and specificity amongst the studies (0.366 to 0.940 and 0.571 to 0.952, respectively).

#### 3.5.2. DJ-1

DJ-1 was reported by two studies (Table 14) [80,81]. The summary weighted AUC of DJ-1 was excellent at 0.925 (95% CI 0.884 to 0.965, *p* < 0.001) (Figure 8). There was a substantial heterogeneity between studies (Q = 3.346, I^2^ = 70.11%; 95%CI: 0.00–93.28). 

#### 3.5.3. Granulocyte-Colony Stimulating Factor (G-CSF)

G-CSF was reported by two studies (Table 15) [46,50]. The summary weighted AUC for G-CSF was acceptable at 0.687 (95% CI 0.610to 0.765, *p* < 0.001) (Figure 8). There was no significant heterogeneity between studies (Q = 0.8143, I^2^ = 0.00%; 95%CI: 0.00–0.00). 

### 3.6. Metabolomic Biomarkers 

There were 4 studies reporting the diagnostic accuracy of 23 different metabolomic biomarkers in a total of 669 patients of which 190 had EC and the remainder were non-cancerous control group patients. All biomarkers were obtained by blood sample. The control groups comprised of healthy women with normal endometrium as well as those with benign endometrial lesions such as polyps and fibroids. Two out of four studies reported findings on endometroid adenocarcinoma only. One study did not specify the histological subtype of EC. The summary of the diagnostic accuracy and performance of those biomarkers is outlined in Table 16.

### 3.7. Circulating Tumor Related Material Biomarkers 

There were three studies reporting on circulating tumor related materials. Two studies reported the diagnostic accuracy of circulating cell-free DNA (cCFDNA) and one study reported on Survivin expressing circulating tumor cells (CTC). All biomarkers were obtained by blood sample. No studies were eligible for meta-analysis. The performance of these biomarkers is reported in Table 17. 

## 4. Discussion

Endometrial cancer is one of the most common malignant tumors in females, and the primary symptom is abnormal vaginal bleeding or discharge. The frequency with which this symptom occurs and the invasive nature of endometrial biopsy, means that at present the triage of women with suspected EC is suboptimal for women and clinicians alike. The optimum biomarker for EC would have a high sensitivity and specificity for detecting EC compared to benign and healthy controls. It would be utilizable for both pre-menopausal and post-menopausal women. Those women at low risk of disease could be reassured without the need for secondary care interventions such as imaging and biopsy. The ideal receiver-operating characteristic area under curve (AUC) would be close to 1, with a minimum of 0.7 to indicate clinical utility as a biomarker. An accurate diagnostic biomarker utilized in primary care could reduce the number of women referred for painful and costly investigations and indeed might also be used for consideration of screening of high-risk groups such as women with Lynch Syndrome or those with multiple risk factors. Hence, the aim of this study was to determine which biomarkers have been assessed for their diagnostic accuracy to date. 

At present, the biomarkers most assessed for their diagnostic accuracy of EC are serum CA125 and HE4 which have been reviewed by multiple studies and meta-analyses. The performance of CA 125 is poor in terms of sensitivity and specificity making it unsuitable for use [20,21]. HE4 may have utility as a diagnostic tool, however it has been assessed by several meta-analyses and each of these point to considerable heterogeneity within the data [21,86,87]. Similarities in the data seem to suggest that HE4 may have high specificity but a lower sensitivity than would be needed as a primary care screening tool. 

There have been promising results yielded from studies assessing biomarker panels and specifically those using spectroscopic techniques [88,89]. The study by Paraskevaidi et al. assessed the diagnostic accuracy of infrared spectroscopy as a method of detecting EC with an overall diagnostic accuracy of 0.83 [88]. Spectroscopic techniques do not allow for analysis of a single biomarker because peaks may be formed by multiple biological entities. As such, these studies were not eligible for inclusion into this review; however, they are simple techniques yielding promising results.

This study identified 35 proteomic biomarkers eligible for inclusion, 34 of which were serum or plasma based and only one was urine based [64]. These included cancer antigens, hormonal proteins, adipokines, angiogenic growth factors and other proteins. Cancer antigens CA15-3, 19-9, 72-4 and CEA were included in the systematic review and meta-analysis [45,48,54,55,67,70,71,72,73]. Overall, they displayed poor diagnostic performance with summary weighted AUC scores of 0.608–0.666, making these unsuitable for use as diagnostic biomarkers for EC. 

Obesity is accompanied by changes in expression of adipose factors that act both locally and systemically. With the known link between obesity, insulin resistance and EC, adipokines and hormones such as leptin have become the focus of intense investigation. In EC, leptin activates STAT3 proteins, which increase their activity in the process of oncogenesis by stimulating proliferation, promoting angiogenesis and avoiding the control of the immune system. There is a positive correlation between leptin levels and body mass index (BMI). Adiponectin is predominantly secreted by visceral adipose tissue and is the most abundant adipokine, with circulating concentrations inversely correlated with adiposity. This review included three studies of leptin and one study of adiponectin [50,51,53,61]. The diagnostic accuracy of leptin is moderate with a summary ROC of 0.757. Adiponectin assessed in one study in tandem with Visfatin showed that adiponectin was inversely associated with EC risk [61]. The AUC of this adiponectin alone was 0.814 and the adiponectin to Visfatin ratio 0.838 suggesting some potential as diagnostic biomarkers. However, similarly to other hormones such as FSH, LH, estradiol and prolactin which are also raised markedly in EC patients, these may be markers of the major risk factors, obesity and polycystic ovarian syndrome rather than EC itself, thus showing poorer specificity than required of a screening biomarker. A meta-analysis of the effect of adipokines in obesity driven cancers using mendelian randomization supports this conclusion as they failed to find any causal role between adipocytokines and EC and other obesity driven cancers [90]. 

G-CSF, YKL-40 and DJ-1, are other markers that have been implicated as biomarkers for the detection of EC [46,48,50,72,76,77,78,79,80,81]. Findings from this review suggest these markers perform well. YKL-40 is poorly understood but appears to be involved in extracellular matrix remodeling and angiogenesis, promoting cell proliferation, migration, differentiation, and tissue remodeling processes during cellular responses to inflammation [76,77,78,79]. The summary weighted AUC for DJ-1 is 0.786, (95%CI 0.667–0.848), however there is high heterogeneity of 79%. 

Exosomes represent a wide group of membrane-bound lipid particles that originate from the plasma membrane or the endosomal system and are secreted from cells. Exosomes released from both healthy and cancer cells, are abundant in body fluids and mediate cell-to-cell communication by shuttling DNA, RNA, lipids, metabolites, and proteins. In this way, exosomes are implicated in numerous physiological processes but also participate in the formation of the tumor microenvironment and cancer progression. Of all the biomolecules contained in exosomes, miRNAs seem to have the most clinical utility in EC diagnosis. MiRNAs are small non-coding single stranded molecules that regulate all hallmarks of cancer as defined by Hanahan and Weinberg, including proliferation, invasion, angiogenesis, as well as influencing cancer cells chemosensitivity [91].

With regard to the review of exosomal biomarkers, 15 studies were eligible for inclusion, and these reported on 29 different exosomal biomarkers and a total of 5527 patients. Amongst this group 2530 had a diagnosis of EC and 2456 were non-cancerous control-group patients [29,30,31,32,33,34,35,36,37,38,39,40,41,42,43]. Only Mi-RNA 21, 27a, and 223 were suitable for meta-analysis as they met the inclusion criteria and were reported by two or more studies. The summary ROC score was 0.825 (95% CI 0.735–0.915), 0.925 (95%CI 0.801–1.00), and 0.813 (95%CI 0.735–0.890) for MiRNA-21, MiRNA 27a and MiRNA-223, respectively. In comparison to HE4, the relatively high performance of MiRNAs is promising. However, it must be noted that MiRNA 27a included in a study by Ghazala et al. reported an AUC of 1.00 making it the perfect test in terms of sensitivity and specificity, positive and negative likelihood ratio [33]. This has not been replicated and has likely overestimated the performance of this biomarker on meta-analysis. The only other study by Wang et al., reported much lower AUC of 0.813 [32]. This may potentially be explained by the fact that Ghazala et al. used a healthy control cohort whereas Wang included a mixed cohort of those women with PMB who had both benign disease and normal endometrium. Both cohorts were relatively small in size with under 75 patients in each study and thus results must be interpreted with caution. 

To the best of our knowledge this is the first study to report and meta-analyze the diagnostic accuracy of different classes sole biomarkers obtained by non-invasive biopsy for detection of EC and thus it may provide a reference guide for those wishing to explore candidate biomarkers for research or those wishing to assess the current evidence for implementation into current practice. 

However, there are numerous limitations to this review and the evidence summarized within it. The main limitation is that there were a low number of small, heterogeneous studies for the majority of the evaluated index tests. It was not apparent from many studies whether a ‘test-phase’ had been conducted prior to the biomarker validation phase which is the point at which the diagnostic accuracy should be assessed. This may undermine the reliability of the summary estimates from the meta-analyses and is likely to have contributed to the marked variability in AUC and sensitivity and specificity seen for most index tests. For the vast majority of biomarkers, no meta-analysis was possible. The decision to meta-analyze using two studies or more was conducted based on numbers of study participants in each study and not solely on the number of studies conducted. However, it is evident that there was significant heterogeneity amongst the biomarkers analyzed. Formal assessments of heterogeneity, such as Eggers’s test were deemed likely too unreliable given the low number of studies in most evaluations.

A significant limitation of this review was also that the studies varied with respect to the control group used, the type, stage and grade of endometrial cancer, the age of the cohorts assessed and the cut-off thresholds for index tests. Additionally, most of the included studies evaluated the diagnostic cut-off thresholds using a ROC analysis without any subsequent validation in an independent cohort. Lack of validation of the diagnostic data in conjunction with the low number of studies for the majority of the presented tests contributed to the low quality of evidence presented in this review. A standardized methodology for fluid biospecimen collection, processing and storage was published in 2014 and would likely improve the quality of studies if adopted for use by future studies. 

The variation in the selection of the case and control groups with inclusion of participants that may not be reflective of the EC population is also a limitation in this systematic review. The recent change towards molecular classification has shifted clinicians away from describing EC cell types as just serous, endometrioid or other histological cell types however some studies made no attempt to sub-classify their EC population other than to report grade and stage. Subgroup analysis of data was considered but due to the size of the studies and relative paucity of studies discussing the same biomarker this would have yielded no more reliable information.

Most studies do not report whether they attempted to reduce selection bias by consecutively enrolling participants. More than two-thirds of the included studies (38/56, 68%) had a two-gate design and included a wide group of participants who underwent surgery for various indications. Inclusion of healthy asymptomatic individuals or participants with other pathological conditions represents a potential selection bias with regard to the control group, which could have biased the test outcomes. The majority of studies involved a normal healthy control group which was either age-matched or in some cases non-age matched. This is likely to overestimate diagnostic accuracy of a biomarker more than in studies where a one-gate design is used and patients with a presenting symptom such as PMB go on to be part of the case or control group. 

We suggest cautious interpretation of the presented results. Although studies demonstrated diagnostic potential for a number of tests, the level of heterogeneity, wide confidence intervals and risk of bias in many studies included in this review undermine reliability of the presented results, and hence these data are insufficient to confidently inform clinical practice at this stage. Additional biomarkers, reported in individual studies, displayed diagnostic estimates that qualified for either replacement or triage tests; however, there were not enough data for a meaningful recommendation on the use of any of these tests. As with all biomarkers and index tests, the next phase of validation will be to assess diagnostic accuracy amongst a non-symptomatic, low risk, low prevalence cohort in order to assess its performance as a true screening test.

## 5. Conclusions

In conclusion this systematic review and meta-analysis has sought to summarize the existing literature on the performance of non-invasive biomarkers for suspected EC triage and diagnosis. There is a clear need for a biomarker with a low false positive rate that can be used in primary care to reduce the number of unwarranted, invasive investigations. It has become clear from review that many biomarkers are still at discovery phase, rather than validation phase and are thus not at a stage where diagnostic accuracy should be assessed. There is also a high degree of heterogeneity between studies, this is likely due to studies reporting on different types of EC. Given that the dualistic Bohkman classification of type-1 and type-2 EC fails to adequately differentiate between EC or provide useful estimates of prognosis, it is likely studies reporting biomarker performance stratified by EC cancer subtype will demonstrate different biomarkers detecting different subtypes of EC. At the very least with more literature available, subgroup meta-analysis will become possible. Furthermore, it is clear from the review of current literature that two-gate study design may inflate the diagnostic accuracy of the biomarkers studied. Healthy control group patients are not reflective of the patient population undergoing evaluation of suspected EC. Until there is a diagnostic biomarker identified that is likely to be capable of high-performance triage amongst the symptomatic population, studies at the biomarker discovery or validation phase should consider inclusion of both healthy and symptomatic control cohorts. 

There is wide recognition that an accurate non-invasive test for EC triage is likely to confer several advantages over the current standard of ultrasound scan and endometrial biopsy. These potential advantages include a reduction in cost (both in direct medical costs and in time off work), reduced discomfort, shorter recovery times and a reduction in the rate of serious complications associated with surgery. Another benefit of an accurate, non-invasive diagnostic test for EC is the prospect of early diagnosis and timely therapeutic interventions to minimize disease progression. Whilst this review highlights several methodological issues with the current body of evidence, there are some promising findings; exosomal compounds, in particular MiRNAs have shown moderate to good performance in the limited available data, but perhaps more reassuringly less heterogeneity between studies. Similarly, amongst proteomic compounds serum YKL-40 and DJ-1 have good to excellent performance and a next step might be to consider larger scale prospective evaluation of these biomarkers in order to determine their utility. 

## Figures and Tables

**Figure 1 cancers-14-04666-f001:**
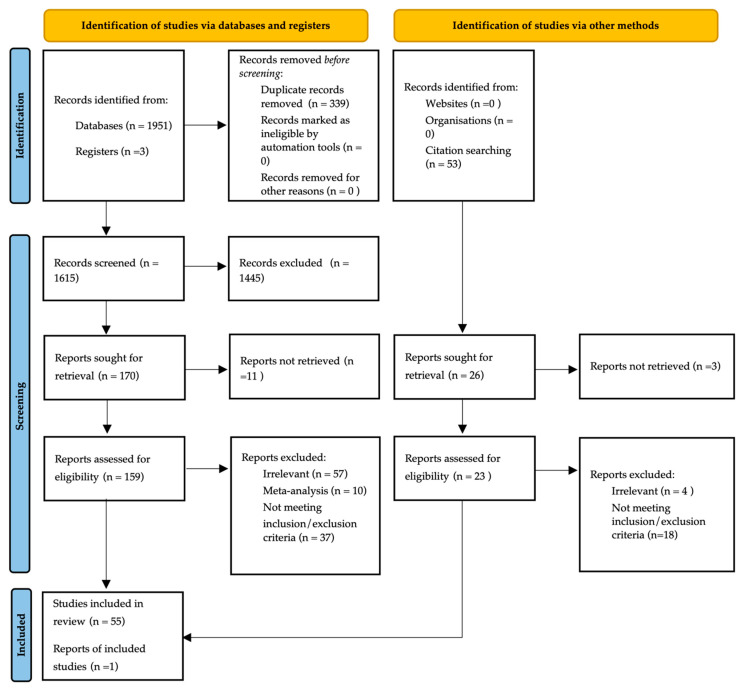
Flowchart of the search strategy.

**Figure 2 cancers-14-04666-f002:**
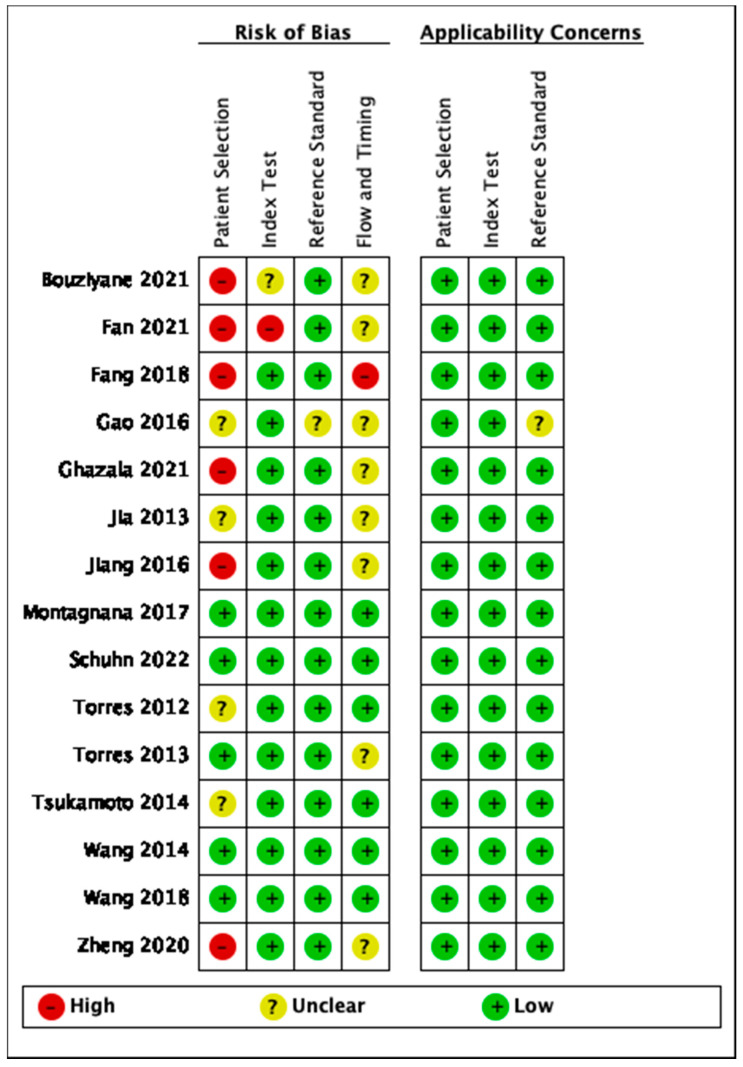
Risk of bias and applicability concerns summary: review authors’ judgements about each domain for exosomal studies.

**Figure 3 cancers-14-04666-f003:**
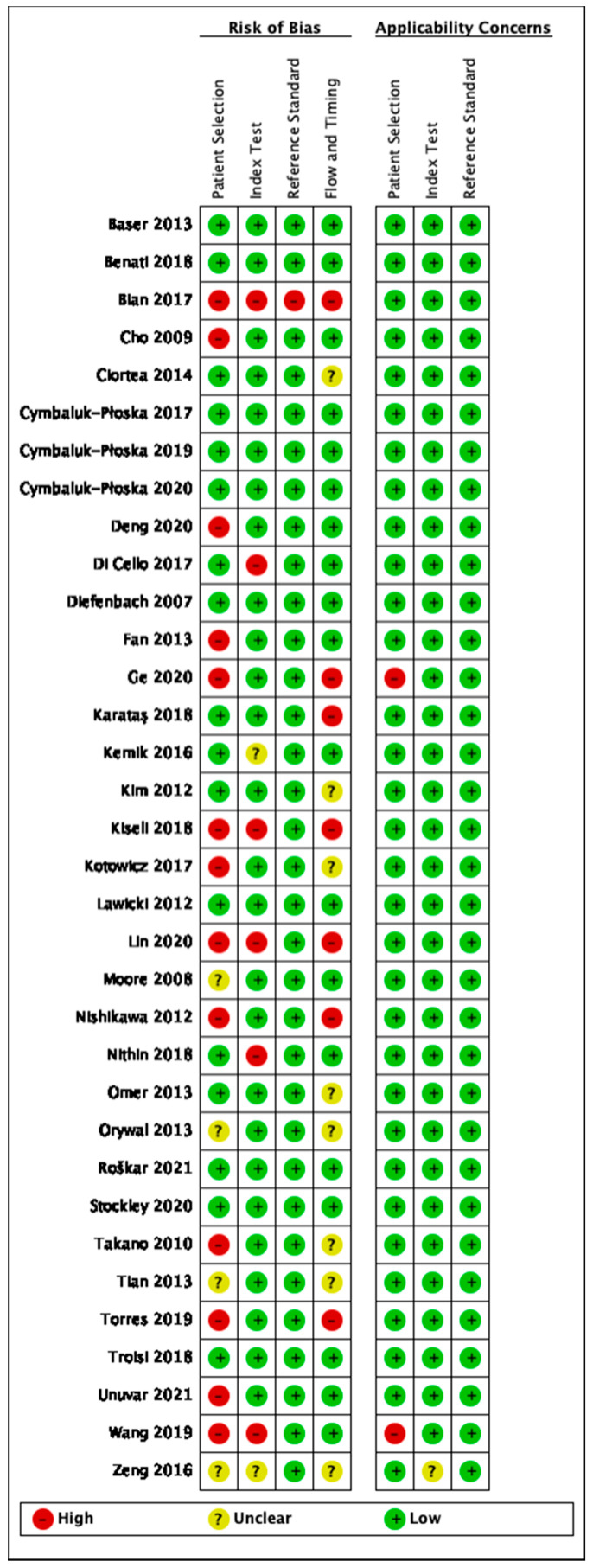
Risk of bias and applicability concerns summary: review authors’ judgements about each domain for proteomic studies.

**Figure 4 cancers-14-04666-f004:**
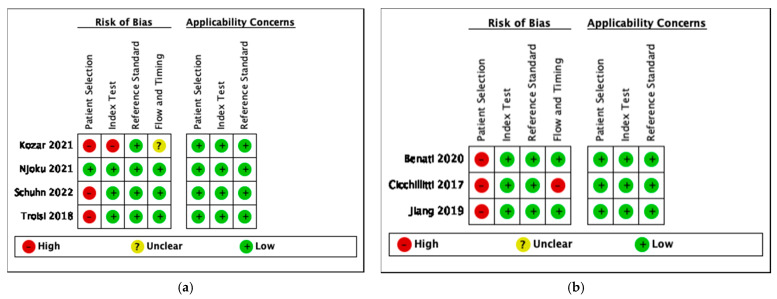
(**a**) Risk of bias and applicability concerns summary: review authors’ judgements about each domain for metabolomic studies (**b**) Risk of bias and applicability concerns summary: review authors’ judgements about each domain for circulating tumor material studies.

**Figure 5 cancers-14-04666-f005:**
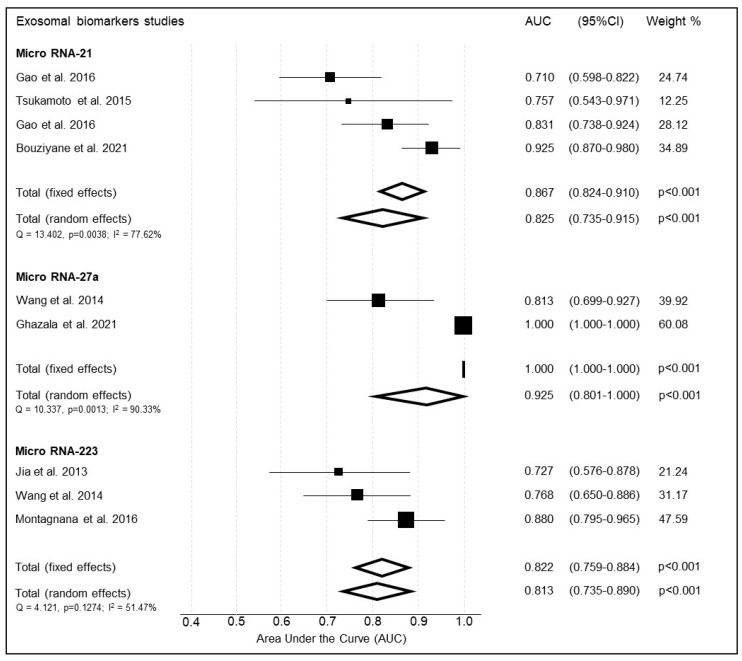
Summary of the diagnostic accuracy of micro-RNAs included in the meta-analysis.

**Figure 6 cancers-14-04666-f006:**
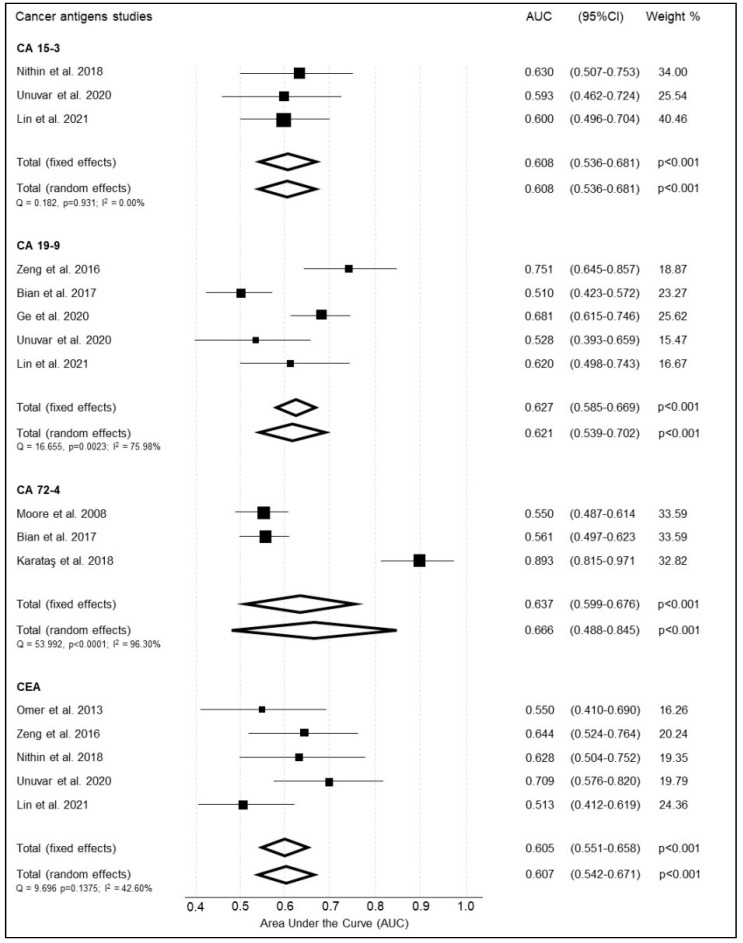
Summary of the diagnostic accuracy of proteomic cancer antigens included in the meta-analysis.

**Figure 7 cancers-14-04666-f007:**
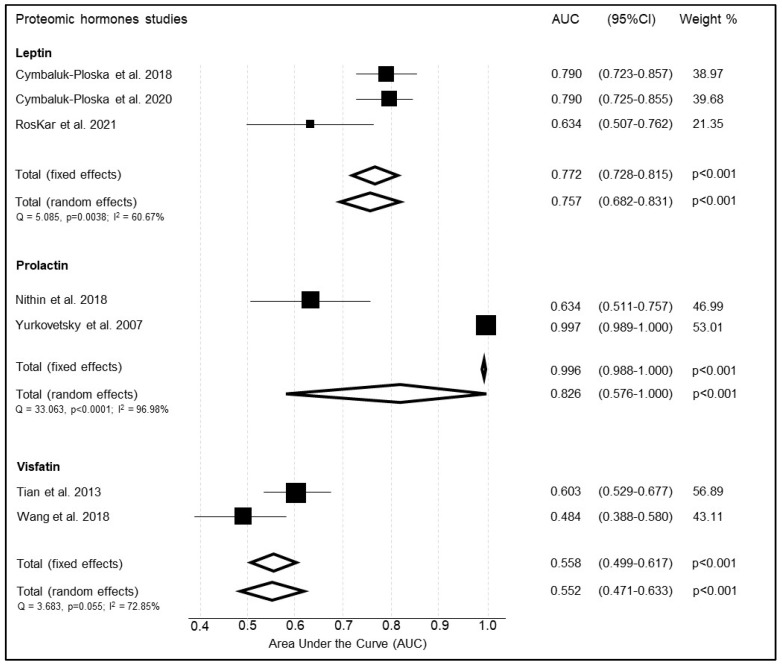
Summary of the diagnostic accuracy of proteomic hormones included in the meta-analysis.

**Figure 8 cancers-14-04666-f008:**
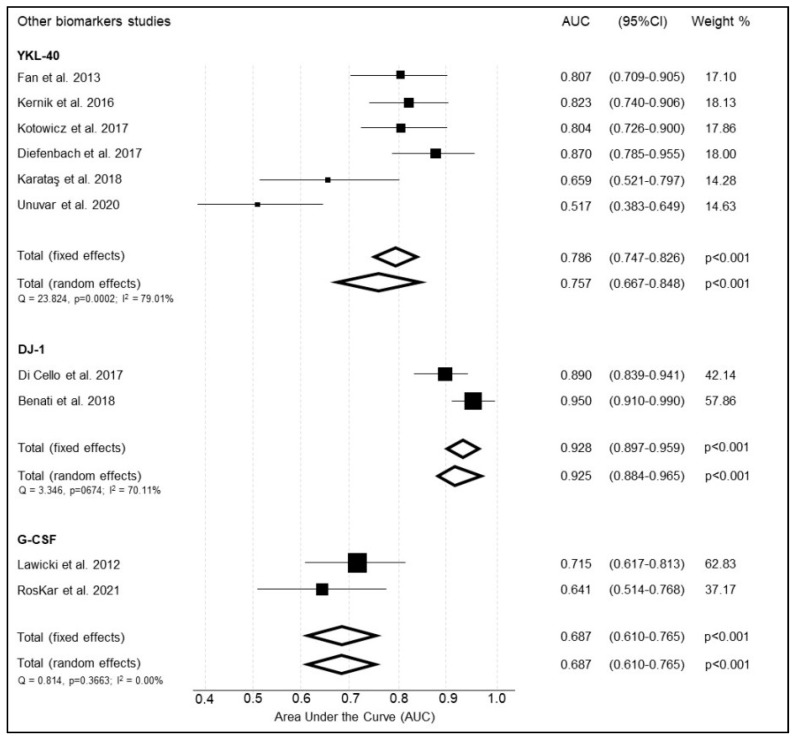
Summary of the diagnostic accuracy of other proteomic biomarkers included in the meta-analysis.

**Table 1 cancers-14-04666-t001:** Summary table of performance of Micro-RNAs for detection of endometrial cancer.

Authors	m-RNA	Cases (n)	Controls (n)	AUC	95% AUC	Sensitivity	Specificity
Acceptable performance							
Fan et al., 2021 ^1^ [36]	484	92	102	0.644	0.566 to 0.722	-	-
Fan et al., 2021 ^1^ [36]	204-5p	92	102	0.668	0.592 to 0.743	-	-
Fan par. 2021 ^1^ [36]	195-5p	92	102	0.669	0.593 to 0.745	-	-
Fan et al., 2021 ^1^ [36]	143-3p	92	102	0.677	0.602 to 0.751	-	-
Fan et al., 2021 ^1^ [36]	423-3p	92	102	0.689	0.611 to 0.767	-	-
Montagnana et al., 2016 ^1^ [35]	186	46	28	0.700	0.580 to 0.830	-	-
Good performance							
Montagnana et al., 2016 ^1^ [35]	222	46	28	0.720	0.590 to 0.850	-	-
Jiang et al., 2016 ^1^ [37]	887-5p	20	20	0.728	0.563 to 0.892	0.950	0.600
Jia et al., 2013 ^1^ [34]	204	26	22	0.740	0.594 to 0.885	-	-
Schuhn et al., 2022 ^1^ [38]	200c	20	157	0.740	0.666 to 0.815	1.000	0.573
Torres et al., 2012 ^1^ [39]	100	34	14	0.740	0.592 to 0.897	0.640	0.790
Fan et al., 2021 ^1^ [36]	20b-5p	92	102	0.756	0.689 to 0.823	-	-
Wang et al., 2014 ^1,2^ [32]	15b	31	33	0.767	0.653 to 0.882	0.740	0.697
Schuhn et al., 2022 ^1^ [38]	320b	20	157	0.774	0.702 to 0.845	0.950	0.659
Schuhn et al., 2022 ^1^ [38]	652	20	157	0.775	0.651 to 0.859	0.900	0.598
Fang et al., 2018 ^1^ [40]	93	176	100	0.781	0.724 to 0.842	-	-
Torres et al., 2012 ^1^ [39]	199b	34	14	0.786	0.642 to 0.892	0.790	0.710
Schuhn et al., 2022 ^1^ [38]	375	20	157	0.796	0.712 to 0.880	0.850	68.700
Excellent performance							
Torres et al., 2012 ^1^ [39]	99a	34	14	0.810	0.669 to 0.909	0.760	0.790
Tsukamoto et al., 2015 ^1^ [30]	30a-3p	28	28	0.813	0.638 to 0.987	-	-
Jia et al., 2013 ^1^ [34]	222	26	22	0.837	0.726 to 0.948	-	-
Jia et al., 2013 ^1^ [34]	186	26	22	0.865	0.755 to 0.974	-	-
Torres et al., 2013 ^1^ [41]	449a	34	14	0.879	0.814 to 0.943	-	-
Torres et al., 2013 ^1^ [41]	1228	34	14	0.890	0.829 to 0.951	-	-
Outstanding performance							
Tsukamoto et al., 2015 ^1^ [30]	135b	28	28	0.972	0.913 to 1.00	-	-
Wang et al., 2018 ^2^ [42]	29-b	356	149	0.976	0.951 to 1.00	0.960	0.979
Ghazala et al., 2021 ^1^ [21]	150-5p	36	36	0.982	0.955 to 1.00	0.890	1.000
Zheng et al., 2019 ^1,2^ [43]	93	100	100	0.990	0.976 to 1.00	0.930	0.970
Montagnana et al., 2016 ^1^ [35]	204	46	28	1.000	-	-	-
Tsukamoto et al., 2015 ^1^ [30]	205	28	28	1.000	-	-	-

^1^ Controls with normal endometrium. ^2^ Controls with benign lesions (polyps). AUC: Area Under the Curve; CI: Confidence Interval.

**Table 2 cancers-14-04666-t002:** Summary table of the performance of Micro-RNA 21.

Author	Cases (n)	Controls (n)	AUC	AUC 95%CI	Sensitivity	Specificity	PPV	NPV
Gao et al., 2016 [29]	50	50	0.710	0.598 to 0.822	0.640	0.760	-	-
Tsukamoto et al., 2015 [30]	12	12	0.757	0.543 to 0.971	-	-	-	-
Gao et al., 2016 [29]	50	50	0.831	0.738 to 0.924	0.700	0.920	-	-
Bouziyane et al., 2021 [31]	71	54	0.925	0.870 to 0.980	0.850	0.868	-	-

AUC: Area Under the Curve; CI: Confidence Interval; PPV: positive predictive value; NPV: negative predictive value.

**Table 3 cancers-14-04666-t003:** Summary table of the performance of Micro-RNA 27a.

Author	Cases (n)	Controls (n)	AUC	AUC 95%CI	Sensitivity	Specificity	PPV	NPV
Wang et al., 2014 [32]	31	33	0.813	0.699 to 0.927	0.770	0.818	-	-
Ghazala et al., 2021 [33]	36	36	1.000	1.000 to 1.000	1.000	1.000	-	-

AUC: Area Under the Curve; CI: Confidence Interval; PPV: positive predictive value; NPV: negative predictive value.

**Table 4 cancers-14-04666-t004:** Summary table of the performance of Micro-RNA 223.

Author	Cases (n)	Controls (n)	AUC	AUC 95%CI	Sensitivity	Specificity	PPV	NPV
Jia et al., 2013 [34]	26	22	0.727	0.576–0.878	0.084	-	-	-
Wang et al., 2014 [32]	31	33	0.768	0.650–0.886	0.065	0.650	0.818	-
Montagnana et al., 2016 [35]	74	28	0.880	0.795–0.965	0.043	-	-	-

AUC: Area Under the Curve; CI: Confidence Interval; PPV: positive predictive value; NPV: negative predictive value.

**Table 5 cancers-14-04666-t005:** Summary table of the performance of all proteins not eligible for inclusion in meta-analysis.

Author	Biomarker	Cases (n)	Controls (n)	AUC	AUC 95%CI	Sensitivity	Specificity	PPV	NPV
Poor performance									
Lin et al., 2021 [44]	AFP	101	475	0.490	0.385–0.594	0.710	0.345	-	-
Moore et al., 2008 ^1^ [45]	SMRP	156	171	0.505	0.443–0.568	-	-	-	-
Lin et al., 2021 [44]	SCC-Ag	101	475	0.512	0.407–0.617	0.903	0.208	-	-
Lawicki et al., 2012 ^1,2^ [46]	IL-3	65	40	0.527	0.413–0.641	0.800	0.980	0.830	0.430
Kim et al., 2012 ^1^ [47]	NLR	238	596	0.539	0.495–0.583	-	0.512	0.591	-
Lawicki et al., 2012 ^1,2^ [46]	GM-CSF	65	40	0.557	0.445–0.669	0.140	0.930	0.750	0.430
Unuvar et al., 2020 [48]	TNC	38	21	0.575	0.440–0.703	0.605	0.619	0.742	0.464
Acceptable performance									
Orywal et al., 2013 [49]	Total ADH	40	52	0.623	0.507–0.739	0.690	0.770	0.620	0.610
Unuvar et al., 2020 [48]	Neopterin	38	21	0.633	0.498–0.755	0.447	0.857	0.850	0.462
Kim et al., 2012 ^1^ [47]	Neutrophil	238	596	0.641	0.598–0.684	-	0.794	0.237	-
RosKar et al., 2021 [50]	Tie-2	36	36	0.652	0.525–0.779	-	-	-	-
Cymbaluk-Ploska et al., 2020 [51]	FGF23	98	84	0.660	0.582–0.738	-	-	-	-
Torres et al., 2019 [52]	EpCAM	45	20	0.667	0.540–0.780	0.420	0.950	0.021	0.998
Unuvar et al., 2020 [48]	Periostin	38	21	0.668	0.533–0.785	0.526	0.857	0.870	0.500
Cymbaluk-Ploska et al., 2018 [53]	Galectin-3	92	76	0.680	0.600–0.760	0.670	0.700	-	-
Orywal et al., 2013 ^1,2^ [49]	ADH1	40	52	0.682	0.570–0.793	0.600	0.630	-	-
Kim et al., 2012 ^1^ [47]	MNM	238	596	0.696	0.655–0.737	-	0.629	0.691	-
Ge et al., 2020 [54]	Fibrinogen	127	96	0.690	0.625–0.724	0.925	0.244	-	-
Good performance									
Lin et al., 2020 [44]	GP6	94	112	0.700	0.630–0.770	-	-	-	-
Kim et al., 2012^1^ [47]	Monocyte	238	596	0.706	0.665–0.747	-	0.550	0.773	-
Ge et al., 2020 [54]	Fibrinogen	127	96	0.717	0.654–0.779	0.945	0.346	-	-
Lin et al., 2020 [44]	GP4	94	112	0.720	0.650–0.790	-	-	-	-
Lin et al., 2020 [44]	GP12	94	112	0.730	0.660–0.800	-	-	-	-
Omer et al., 2013 [49]	SAA	64	34	0.730	0.600–0.860	0.687	0.586	0.786	0.459
Unuvar et al., 2020 [48]	IDO	38	21	0.733	0.602–0.840	0.868	0.571	0.786	0.706
Lin et al., 2020 [55]	GP14	94	112	0.740	0.680–0.810	-	-	-	-
Lawicki et al., 2012 ^1,2^ [46]	SCF	65	40	0.751	0.659–0.843	0.430	0.930	0.900	0.530
Cho et al., 2009^1^ [56]	Osteopontin	56	154	0.758	0.678–0.838	0.627	0.779	-	-
Cymbaluk-Ploska et al., 2019 [57]	Lipocalin-2	52	67	0.760	0.660–0.850	0.840	0.780	-	-
Kiseli et al., 2018 [58]	pro-GRP	37	32	0.775	0.667–0.882	0.607	0.814	0.680	0.761
Cymbaluk-Ploska et al., 2017 [59]	MMP2	62	50	0.790	0.707–0.873	0.680	0.860	-	-
Lawicki et al., 2012 ^1,2^ [46]	M-CSF	65	40	0.794	0.710–0.878	0.690	0.930	0.940	0.680
Nishikawa et al., 2012 ^1^ [60]	GRO alpha	39	38	0.799	0.699–0.899	-	-	-	-
Excellent performance									
Cymbaluk-Ploska et al., 2020 [51]	FGF21	98	84	0.810	0.748–0.872	-	-	-	-
Wang et al., 2019 [61]	Adiponectin	53	98	0.814	0.747–0.881	0.857	0.726	-	-
Cymbaluk-Ploska et al., 2018 [53]	Omentin-1	92	76	0.820	0.678–0.838	0.850	0.790	-	-
Baser et al., 2013 ^2^ [62]	SPAG9	63	37	0.820	0.739–0.901	0.740	0.830	0.880	0.645
Jiang et al., 2019 [63]	TOPO48	80	80	0.826	0.743–0.913	-	-	-	-
Stockley et al., 2020 [64]	MCM5 *	41	58	0.830	0.740–0.920	0.878	0.759	-	-
Torres et al., 2019 [52]	CD44	45	20	0.834	0.710–0.920	0.490	1.000	1.000	0.998
Takano et al., 2010 ^1^ [65]	m/z 28000	40	40	0.860	0.777–0.943	0.943	-	-	-
Cymbaluk-Ploska et al., 2018 [53]	Vaspin	92	76	0.860	0.804–0.916	0.890	0.830	-	-
Takano et al., 2010 ^1^ [65]	m/z 6680	40	40	0.880	0.803–0.957	-	-	-	-
Takano et al., 2010 ^1^ [65]	m/z 9300	40	40	0.880	0.039–0.803	0.957	-	-	-
Deng et al., 2020 [66]	COX2	61	32	0.887	0.822–0.952	0.951	0.719	-	-
Outstanding performance									
Torres et al., 2019 [52]	TGM2	45	20	0.901	0.790–0.970	0.780	1.000	1.000	0.999
Takano et al., 2010 ^1^ [65]	m/z 3340	40	40	0.920	0.032–0.857	0.983	-	-	-
Zeng et al., 2016 [67]	IL-33	160	160	0.929	0.860–0.998	-	-	-	-
Deng et al., 2020 [66]	wnt3a	61	32	0.931	0.881–0.981	0.967	0.812	-	-
Ciortea et al., 2014 ^1^ [68]	IL-8	44	44	0.940	0.888–0.992	-	-	-	-
Troisi et al., 2017 ^1^ [69]	Progesterone	88	80	0.965	0.925–1.000	-	-	-	-
Zeng et al., 2016 ^1^ [67]	IL-31	160	160	0.973	0.945–0.998	-	-	-	-
Troisi et al., 2017 ^1^ [69]	Lactic Acid	88	80	1.000	-	-	-	-	-

^1^ Controls with normal endometrium. ^2^ Controls with benign lesions (polyps). * Urine derived biomarker. AUC: Area Under the Curve; CI: Confidence Interval; PPV: positive predictive value; NPV: negative predictive value.

**Table 6 cancers-14-04666-t006:** Summary table of the performance of CA 15-3.

Author	Cases (n)	Controls (n)	AUC	AUC 95%CI	Sensitivity	Specificity	PPV	NPV
Nithin et al., 2018 [70]	38	40	0.630	0.506–0.754	0.447	0.825	0.708	0.611
Unuvar et al., 2020 [48]	38	21	0.593	0.457–0.719	0.526	0.714	0.769	0.455
Lin et al., 2020 [55]	101	475	0.600	0.496–0.705	0.613	0.593	-	-

AUC: Area Under the Curve; CI: Confidence Interval; PPV: positive predictive value; NPV: negative predictive value.

**Table 7 cancers-14-04666-t007:** Summary table of the performance of CA 19-9.

Author	Cases (n)	Controls (n)	AUC	AUC 95%CI	Sensitivity	Specificity	PPV	NPV
Zeng et al., 2016 [67]	160	160	0.751	0.645–0.857	0.813	0.479	-	-
Bian et al., 2017 ^1^ [71]	105	87	0.510	0.423–0.572	0.163	-	0.510	0.590
Ge et al., 2020 [54]	96	31	0.681	0.615–0.746	0.945	0.047	-	-
Unuvar et al., 2020 [48]	38	21	0.528	0.393–0.659	0.290	1.000	1.000	0.438
Lin et al., 2020 [55]	101	475	0.620	0.498–0.743	0.548	0.747	-	-

AUC: Area Under the Curve; CI: Confidence Interval; PPV: positive predictive value; NPV: negative predictive value. ^1^ Wilcoxon statistics used where no 95% CI reported.

**Table 8 cancers-14-04666-t008:** Summary table of the performance of CA-72-4.

Author	Cases (n)	Controls (n)	AUC	AUC 95%CI	Sensitivity	Specificity	PPV	NPV
Moore et al., 2008 [45]	156	171	0.550	0.487–0.614	-	-	-	-
Bian et al., 2017 [71]	105	87	0.561	0.497–0.623	0.113	-	0.500	0.650
Karataş et al., 2018 [72]	41	21	0.893	0.815–0.971	0.976	0.714	-	-

AUC: Area Under the Curve; CI: Confidence Interval; PPV: positive predictive value; NPV: negative predictive value.

**Table 9 cancers-14-04666-t009:** Summary table of the performance of CEA.

Author	Cases (n)	Controls (n)	AUC	AUC 95%CI	Sensitivity	Specificity	PPV	NPV
Omer et al., 2013 [73]	64	34	0.550	0.410–0.690	0.587	0.427	0.698	0.316
Zeng et al., 2016 [67]	160	160	0.644	0.524–0.764	0.800	0.457	-	-
Nithin et al., 2018 [70]	38	40	0.628	0.504–0.752	0.342	0.950	0.867	0.603
Unuvar et al., 2020 [48]	38	21	0.709	0.576–0.820	0.474	0.905	0.900	0.487
Lin et al., 2021 [55]	101	475	0.513	0.412–0.619	0.882	0.236	-	-

AUC: Area Under the Curve; CI: Confidence Interval; PPV: positive predictive value; NPV: negative predictive value.

**Table 10 cancers-14-04666-t010:** Summary table of the performance of Leptin.

Author	Cases (N)	Controls (N)	AUC	AUC 95%CI	Sensitivity	Specificity	PPV	NPV
Cymbaluk-Ploska et al., 2018 [53]	92	76	0.790	0.723–0.857	0.840	0.720	-	-
Cymbaluk-Ploska et al., 2020 [51]	98	84	0.790	0.725–0.855	0.820	0.710	-	-
RosKar et al., 2021 [50]	36	36	0.634	0.506–0.762	-	-	-	-

AUC: Area Under the Curve; CI: Confidence Interval; PPV: positive predictive value; NPV: negative predictive value.

**Table 11 cancers-14-04666-t011:** Summary table of the performance of Prolactin.

Author	Cases (n)	Controls (n)	AUC	AUC 95%CI	Sensitivity	Specificity	PPV	NPV
Nithin et al., 2018 [70]	38	40	0.634	0.510–0.758	0.386	0.875	0.737	0.593
Yurkovetsky et al., 2007 [74]	115	135	0.997	0.990–1.004	0.983	0.980	-	-

AUC: Area Under the Curve; CI: Confidence Interval; PPV: positive predictive value; NPV: negative predictive value.

**Table 12 cancers-14-04666-t012:** Summary table of the performance of Visfatin.

Author	Cases (n)	Controls (n)	AUC	AUC 95%CI	Sensitivity	Specificity	PPV	NPV
Tian et al., 2013 [75]	120	70	0.603	0.528–0.677	0.758	0.567	-	0.542
Wang et al., 2018 [61]	53	98	0.484	0.388–0.579	-	-	-	-

AUC: Area Under the Curve; CI: Confidence Interval; PPV: positive predictive value; NPV: negative predictive value.

**Table 13 cancers-14-04666-t013:** Summary table of the performance of YKL-40.

Author	Cases (N)	Controls (N)	AUC	AUC 95%CI	Sensitivity	Specificity	PPV	NPV
Fan et al., 2013 [76]	50	50	0.807	0.709–0.905	0.735	0.816	0.694	0.844
Kemik et al., 2016 [77]	34	60	0.823	0.740–0.906	0.940	0.480	-	-
Kotowicz et al., 2017 [78]	41	21	0.804	0.726–0.900	0.689	0.800	-	-
Diefenbach et al., 2017 [79]	34	44	0.870	0.785–0.955	0.760	0.930	-	-
Karataş et al., 2018 [72]	74	25	0.659	0.521–0.797	0.366	0.952	0.938	0.435
Unuvar et al., 2020 [48]	38	21	0.517	0.383–0.649	0.605	0.571	0.719	0.444

AUC: Area Under the Curve; CI: Confidence Interval; PPV: positive predictive value; NPV: negative predictive value.

**Table 14 cancers-14-04666-t014:** Summary table of the performance of DJ-1.

Author	Cases (n)	Controls (n)	AUC	AUC 95%CI	Sensitivity	Specificity	PPV	NPV
Di Cello et al., 2017 [80]	101	44	0.890	0.839–0.941	0.753	0.796	0.583	0.894
Benati et al., 2018 [81]	45	29	0.950	0.910–0.990	0.890	0.900	-	-

AUC: Area Under the Curve; CI: Confidence Interval; PPV: positive predictive value; NPV: negative predictive value.

**Table 15 cancers-14-04666-t015:** Summary table of the performance of G-CSF.

Author	Cases (n)	Controls (n)	AUC	AUC 95%CI	Sensitivity	Specificity	PPV	NPV
Lawicki et al., 2012 [46]	65	40	0.715	0.618–0.812	0.210	0.930	0.820	0.450
RosKar et al., 2021 [50]	36	36	0.641	0.513–0.769	-	-	-	-

AUC: Area Under the Curve; CI: Confidence Interval; PPV: positive predictive value; NPV: negative predictive value.

**Table 16 cancers-14-04666-t016:** Summary table of the performance of metabolites not eligible for inclusion in meta-analysis.

Author	Metabolite	Cases (n)	Controls (n)	AUC	AUC 95%CI	Sensitivity	Specificity	PPV	NPV
Good performance									
Kozar et al., 2020 ^4^ [82]	1-Methyladenosine	15	21	0.746	0.576–0.916	0.670	0.810	-	-
Schuhn et al., 2022 ^1^ [38]	One CpG site at at S100P,	20	157	0.750	0.641–0.858	0.895	0.545	-	-
Schuhn et al., 2022 ^1^ [38]	Tetrade-Cenoylcarnitine	20	157	0.751	0.647–0.856	0.800	0.690	-	-
Kozar et al., 2020 ^4^ [82]	AC 16:1-OH	15	21	0.759	0.577–0.941	0.600	0.950	-	-
Kozar et al., 2020 ^4^ [82]	Cer 40:1; 2	15	21	0.768	0.610–0.927	0.670	0.810	-	-
Schuhn et al., 2022 ^1^ [38]	One CpG site at RAPSN	20	157	0.772	0.665–0.889	0.737	0.752	-	-
Schuhn et al., 2022 ^1^ [38]	Carnitine	20	157	0.792	0.710–0.873	0.950	0.579	-	-
Schuhn et al., 2022 ^1^ [38]	Acetylcarnitine	20	157	0.800	0.715–0.884	0.950	0.608	-	-
Excellent performance									
Njoku et al., 2021 ^2^ [83]	3-Hydroxybutyrate	67	69	0.817	0.737–0.884	-	-	-	-
Schuhn et al., 2022 ^1^ [83]	Malonylcarnitine	20	157	0.819	0.721–0.918	0.800	0.731	-	-
Njoku et al., 2021 ^2^ [83]	1-1- Enyl-Stearoyl-2 Oleoyl GPE	67	69	0.825	0.750–0.888	-	-	-	-
Njoku et al., 2021 ^2^ [83]	3-Hydroxy-Butyrlcarnitine	67	69	0.826	0.752–0.853	-	-	-	-
Kozar et al., 2020 ^4^ [82]	Cer 34:1; 2	15	21	0.835	0.705–0.965	0.730	0.810	-	-
Njoku et al., 2021 ^2^ [83]	1-1- Enyl-Stearoyl-GPE	67	69	0.841	0.767–0.900	-	-	-	-
Njoku et al., 2021 ^2^ [83]	1-linolenoyl-GPC	67	69	0.844	0.776–0.909	-	-	-	-
Njoku et al., 2021 ^2^ [83]	1-(1-enyl-stearoyl)-2-linoleoyl-GPE	67	69	0.853	0.780–0.910	-	-	-	-
Outstanding performance									
Njoku et al., 2021 ^2^ [83]	1-Lignoceroyl GPC	67	69	0.910	0.860–0.950	-	-	-	-
Troisi et al., 2018 ^3^ [69]	Stearic Acid	88	80	0.943	0.893–0.979	-	-	-	-
Troisi et al., 2018 ^3^ [69]	Homocysteine	88	80	0.952	0.906–0.989	-	-	-	-
Troisi et al., 2018 ^3^ [69]	Threonine	88	80	0.979	0.933–1.000	-	-	-	-
Troisi et al., 2018 ^3^ [69]	Valine	88	80	0.999	0.995–1.000	-	-	-	-
Troisi et al., 2008 ^3^ [69]	Myristic Acid	88	80	1.000	0.996–1.000	-	-	-	-

^1^ Tested by electrospray ionisation tandem mass spectrometry (ESI–MS/MS). ^2^ Tested by mass spectrometry. ^3^ Tested by gas-chromatography mass-spectrometry. ^4^ Tested by the ultra-performance liquid chromatography coupled with triple-quadruple tandem mass spectrometry (UPLC-TQ/MS). AUC: Area Under the Curve; CI: Confidence Interval; PPV: positive predictive value; NPV: negative predictive value.

**Table 17 cancers-14-04666-t017:** Summary table of the performance of circulating tumor not eligible for inclusion in meta-analysis.

Author	Biomarker	Cases (n)	Controls (n)	AUC	AUC 95%CI	Sensitivity	Specificity	PPV	NPV
Cicchillitti et al., 2017 [84]	cCFDNA	59	21	0.704	0.632–0.777	0.521	0.839	-	-
Jiang et al., 2019 [63]	cCFDNA	80	80	0.791	0.657–0.887	-	-	-	-
Benati et al., 2020 [85]	Survivin-expressing CTC	40	31	0.870	0.790–0.950	0.800	0.807	-	-

AUC: Area Under the Curve; CI: Confidence Interval; PPV: positive predictive value; NPV: negative predictive value.

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
