# Peer review of "Diagnostic Accuracy of Liquid Biomarkers for the Non-Invasive Diagnosis of Endometrial Cancer: A Systematic Review and Meta-Analysis"

_cancers, 2022, doi:10.3390/cancers14194666_

Round 1

Reviewer 1 Report

This is a nice systematic review and meta-analysis about the usefulness of the different biomarkers found in urine and blood to diagnose endometrial cancer. I find the work rigorous and authors are very honest about the findings and the limitations. However, the work needs some revision throughout the text.

Introduction

Line 61: define HRT

Line 89: authors mention “Advances in the field of areas such as genomic, proteomic and metabolomic sequencing”. In my opinion, sequencing only applies for genomics. Proteomics and metabolomics offer protein and metabolite identification through mass spectrometry approaches.

Line 88: authors say “receiver operator characteristic (ROC) curve plotting sensitivity over specificity”. To be correct, it should be sensitivity over the 1-specificity.

 Methods

Line 137: “two independent authors (RK, SW, JC)”. Were they two or three?

Line 174: authors could mention that I2 is the Higgin’s index

Results

Figure 5 and 6: consider to show the funnel plots as they could be helpful to visualize the publication bias

Line 262, 281, 306 : I2 instead of I2

Line 279: CA15-3 weighted AUC should be  0.608 according to forest plot

Line 295: CA72-4 weighted AUC should be 0.666 according to forest plot

Line 305: CEA weighted AUC should be 0.605 according to forest plot

Autors should specify which markers were detected in urine and which in blood. They could consider to add a separate column in tables with this information.

Reviewer 2 Report

There is a fundamental flaw in this paper.  The authors have done a systematic review of biomarkers in blood and urine for endometrial cancer detection.

Unless I have missed it , the results are not stratified into blood and urine .  They appear to be pooled , or there are no urine tests included .

It needs to clearly define whether the included studies are just blood studies or if there are urine studies need need to be highlighted as such.

Round 2

Reviewer 1 Report

I think the paper can be accepted in the present form

Reviewer 2 Report

Thank u for stratifying for blood and urine , it is much clearer now that only one study is in urine. This one study was excluded from the final results. 

I think there needs to be clarification that these studies were all done in high prevalence cohort studies .   Ultimately these biomarkers will need to be done in low prevalence studies 

this study by my team was not included ! Is there a reason for this , it was published in this journal 

Cancers (Basel). 2020 May; 12(5): 1256.  Published online 2020 May 16. doi: 10.3390/cancers12051256 PMCID: PMC7281323 PMID: 32429365

Detecting Endometrial Cancer by Blood Spectroscopy: A Diagnostic Cross-Sectional Study

Maria Paraskevaidi,1,2,* Camilo L. M. Morais,1 Katherine M. Ashton,3 Helen F. Stringfellow,3Rhona J. McVey,4 Neil A. J. Ryan,5 Helena O’Flynn,5 Vanitha N. Sivalingam,5 Sarah J. Kitson,5Michelle L. MacKintosh,6 Abigail E. Derbyshire,6 Cecilia Pow,5 Olivia Raglan,2Kássio M. G. Lima,7 Maria Kyrgiou,2,8 Pierre L. Martin-Hirsch,9, Francis L. Martin,1, and Emma J. Crosbie5,6,

Author Response

We would like the thank the reviewer for these comments. We have now mentioned that the analyses were performed in high prevalence studies and have added the following sentence. "As with all biomarkers and index tests, the next phase of validation will be to assess diagnostic accuracy amongst a non-symptomatic, low risk, low prevalence cohort in order to assess its performance as a true screening test". 

Regarding the suggestion of the manuscript by your team, we apologise for not including it in our review. We assessed the study against our inclusion and exclusion criteria but felt that as spectroscopic techniques fail to identify a specific diagnostic biomarker the study did not meet our criteria for inclusion. It is evident however that there is still exceptional merit to such techniques and as such we have referenced this paper in our own review as an alternative approach. 

We have added the following under discussion: "There have been promising results yielded from studies assessing biomarker panels and specifically those using spectroscopic techniques [89,90]. The study by Paraskevaidi et al. assessed the diagnostic accuracy of infrared spectroscopy as a method of detecting EC with an overall diagnostic accuracy of 0.83 [89]. Spectroscopic techniques do not allow for analysis of a single biomarker because peaks may be formed by multiple biological entities. As such, these studies weren’t eligible for inclusion into this review, however, they are simple techniques yielding promising results.

Round 3

Reviewer 2 Report

I am happy with the paper , should be published